# Functional Mitral Regurgitation in Patients with Preserved Ejection Fraction Qualified for Pulmonary Vein Isolation: A Negative Prognostic Factor for Catheter Ablation Efficacy

**DOI:** 10.3390/medicina57080798

**Published:** 2021-08-03

**Authors:** Małgorzata Cichoń, Maciej Wybraniec, Magdalena Mizia-Szubryt, Katarzyna Mizia-Stec

**Affiliations:** First Department of Cardiology, Medical University of Silesia, 40-635 Katowice, Poland; maciejwybraniec@gmail.com (M.W.); mmizia7@gmail.com (M.M.-S.); kmizia@gmail.com (K.M.-S.)

**Keywords:** functional mitral regurgitation, atrial fibrillation, pulmonary vein isolation, ablation efficacy, sinus rhythm maintenance

## Abstract

*Background and Objectives:* Functional mitral regurgitation (F-MR) observed in patients with atrial fibrillation could affect the effectiveness of the sinus rhythm restoring procedures. The aim of the study was to evaluate the impact of F-MR on pulmonary vein isolation (PVI) efficacy in patient with preserved ejection fraction (EF). *Materials and Methods:* One hundred and thirty-six patients with EF ≥ 50% (65.4% males; mean age 56 ± 11 years) with symptomatic paroxysmal or persistent AF qualified for PVI were enrolled into the study. F-MR assessment was performed in transthoracic (TTE) and transesophageal (TEE) echocardiography before the PVI procedure. PVI efficacy was evaluated in three-month and long-term follow-up. *Results:* F-MR was diagnosed in 74.3% patient in transthoracic echocardiography (TTE) (trace: 26.5%, mild: 43.4%, moderate: 3.7%, severe 0.7%) and 94.9% in transesophageal echocardiography (TEE) (trace: 17.6%, mild: 59.6%, moderate: 16.2%, severe: 1.5%). The PVI three-month efficacy was 75.7% in the three-month and 64% in the long-term observation. Severe F-MR in TEE at baseline was associated with lower three-month PVI efficacy (*p* = 0.012), while moderate to severe F-MR in TEE was related to inefficient PVI assessed in long-term follow-up (*p* = 0.041). *Conclusions:* Significant F-MR confirmed by TEE predicts three-month as well as long-term PVI efficacy.

## 1. Introduction

While secondary mitral regurgitation (MR) resulting from left ventricular (LV) systolic dysfunction constitutes a well-known issue, functional mitral regurgitation (F-MR) in patients with preserved EF requires further research. This form of MR is typical for patients with atrial fibrillation (AF) and/or heart failure with preserved ejection fraction (HFpEF).

In the absence of LV dysfunction, a left atrial (LA) enlargement accompanied by mitral annulus (MA) dilation due to AF lead to F-MR [1]. Additionally, the disruption of the MA saddle shape, reduction of MA contractility, MA area to leaflets area imbalance, presence of atriogenic leaflet tethering, LA dysfunction, and LV remodelling participate in the process of F-MR development [2,3].

Pulmonary vein isolation (PVI) is a well-established method for AF recurrence prevention. Rhythm control treatment with PVI is much more effective than therapy with antiarrhythmic drugs. Furthermore, PVI reduces arrhythmia-related symptoms in patients without sinus rhythm maintenance.

LA size, AF durations, and patient age are confirmed risk factors of AF recurrence after catheter ablation. Previous studies demonstrated that F-MR could also be associated with worse clinical outcomes in patients referred for AF ablation.

Considering the increasing number of individuals with AF and the resultant F-MR, the aim of the study was to evaluate the impact of F-MR on pulmonary vein isolation (PVI) efficacy.

## 2. Materials and Methods

### 2.1. Methods

One hundred and thirty-six patients (47 females, 89 men; mean age: 56 ± 11 years) with paroxysmal or persistent symptomatic AF who were hospitalized in the department of cardiology between 2018 and 2019 and who qualified for percutaneous pulmonary vein isolation (PVI) were prospectively enrolled into the study. The primary endpoint of the study was PVI efficacy defined as sinus rhythm maintenance, assessed three months after the procedure and in a long-term follow-up. Individuals with idiopathic cardiomyopathy, LV systolic dysfunction, or organic mitral regurgitation, or patients who underwent cardiac surgery due to valvular heart disease, were excluded from the study. An informed written consent was obtained from all the patients before inclusion in the study.

### 2.2. Laboratory Testing

All the patients who were submitted to PVI underwent basic laboratory tests, including complete blood count, serum creatine concentration with estimated glomerular filtration rate using the modification of diet in renal disease formula (MDRD), liver function parameters, and clotting time assessment. Blood samples were collected from the antecubital vein on the day of admission to the hospital.

### 2.3. Echocardiography

Precise transthoracic echocardiography (TTE) was performed at baseline by a single experienced investigator using an Epiq 7G (Philips, Andover, MA, USA) with a 2.5-MHz probe in 2D, M, and Doppler modes. The study included measurements of the heart chamber dimensions, myocardial contractility, and the assessment of valvular function, with a special emphasis on mitral valve evaluation.

Every patient included in the study underwent transoesophageal echocardiography (TEE) within 24 h preceding the procedure to exclude the potential presence of an LA appendage thrombus and to assess valvular function.

### 2.4. Definitions

F-MR was defined as MR observed in patient with AF after excluding organic MV disease. According to EACVI guidelines, qualitative and semi-quantitative parameters, including the assessment of MV morphology, color flow MR jet, flow convergence zone, continuous wave signal of MR, vena contracta, presence of systolic pulmonary vein flow reversal, and mitral inflow measurement, were used in MR severity grading [4]. F-MRs with a large defect of coaptation were classified as severe F-MR. F-MRs with a dense, triangular regurgitation jet, large flow convergence zone, vena contracta equal or larger than 7 mm, with systolic pulmonary reversal or dominant E-wave, were classified similarly.

Paroxysmal AF was defined as arrhythmia with the typical pattern of irregular RR intervals and absence of distinct *p* waves observed in ECG lasting less than seven days. Persistent AF was defined as an arrhythmia episode lasting at least seven days and long- standing persistent AF of more than a year [5].

PVI efficacy was obtained in a three-month and long-term period (median: 15 months) and defined as an absence of AF in 12-lead ECG and lack of AF symptoms.

### 2.5. PVI Procedure

Every patient who qualified for the procedure received vitamin K antagonists (VKA) or non-vitamin K antagonist oral anticoagulants (NOACs) for at least three months before PVI. Prothrombin time reflected by international normalized ratio (INR) was assessed systematically in patients treated with VKA in order to provide adequate anticoagulation.

A quadripolar electrode was placed in the right ventricle via the left femoral vein, while circular mapping and radiofrequency ablation electrodes or a cryoablation balloon were introduced via the right femoral vein using the Seldinger technique. Jugular access was used for coronary sinus catheterisation.

After the LA rotational angiography, a trans-septal puncture was performed under the control of an optical scope. Three-dimensional electro-anatomical mapping of left atrium was created using the CARTO^®^3 system (Biosense Webster, Diamond Bar, CA, USA). The patients underwent radiofrequency ablation using a ThermoCool^®^ SmartTouch^®^ SF catheter (Biosense Webster, Diamond Bar, CA, USA) or balloon cryoablation using an Arctic Front Advance™ catheter (Medtronic, Minneapolis, MN, USA) under the guidance of a circular mapping electrode, Lasso (Biosense Webster, Diamond Bar, CA, USA) or Achieve (Medtronic, Minneapolis, MN, USA).

Anticoagulation during the procedure involved an intravenous bolus of unfractionated heparin.

### 2.6. Statistical Analysis

Statistical analysis was calculated using SPSS v.25.0 software (IBM Corp, Armonk, NY, USA) and MedCalc v.14.8.1 software (MedCalc Software, Ostend, Belgium). Continuous variables were expressed as the mean ± standard deviation (SD) or median (1–3 quartile boundary). Qualitative parameters were shown as crude numbers and percentages. The type of continuous variable distribution was acquired using a Shapiro–Wilk test. The inter-group difference in the case of non-normally distributed variables was verified using two-tailed Mann–Whitney U, while, in the case of normal distribution, a student’s *t*-test was utilized. The significance of proportions in contingency tables was calculated using a chi-squared test with Bonferroni adjustment. The receiver operating characteristics curve analysis (ROC) was carried out to derive the applicability of variables for the prediction of AF recurrence. The variables with *p* < 0.1 in univariate analysis were incorporated into a Cox proportional hazards model to determine independent predictors of AF recurrence. The universal *p*-value level < 0.05 was regarded as statistically significant throughout the analyses.

## 3. Results

### 3.1. Demographic and Clinical Characteristics of Study Population

The study group consisted of 136 individuals (47 females, 89 men) with AF qualified for PVI. The majority of patients had paroxysmal AF (80.9%), while 16.2% of patients had persistent and <3% long-standing persistent AF. More than one-third of the patients were at high risk of arterial thromboembolism (CHA2DS2-VASc ≥ 3 pts). Patients referred for PVI were significantly symptomatic (EHRA 2: 49.3%, EHRA 3: 45.6%, EHRA 4:3.7%). The most prevalent co-morbidities were atrial hypertension (80.2%), diabetes mellitus (16.9%), chronic kidney disease (11%), and coronary artery disease (10.3%). Eight individuals suffered from heart failure with a preserved ejection fraction (HFpEF). Nearly 10% of the patients had already experienced an ischemic stroke or TIA episode, but no one suffered from intracranial bleeding. Rivaroxaban was the most frequently used anticoagulant drug (46% vs. 39% for dabigatran, 6% for acenocoumarin, 5% for apixaban, and 4% for warfarin).

### 3.2. Echocardiographic Assessment

The prevalence of F-MR in whole study group is presented in Table 1. F-MR was observed in the majority of patients who qualified for PVI, regardless of the applied echocardiographic technique (74.3% in TTE and 94.9% TEE). Both in TTE and TEE, mild F-MR was the most commonly observed severity of MV defect (Table 1).

Nearly half of the study group was affected by LV remodeling or LV hypertrophy; 37% of patients had LV hypertrophy (concentric hypertrophy: 7.9%, eccentric hypertrophy: 29.1%), while concentric remodeling of LV occurred in 10.2% of patients.

### 3.3. F-MR Grade Assessed in TEE

The study revealed no differences in the clinical characteristics between patients with the presence of any grade of F-MR and patients without mitral valve disease.

In the comparison of individuals with different F-MR grade, patients with moderate to severe F-MR were characterized by significantly higher serum creatinine concentration (1.06 vs. 0.91 mg/dL, *p* = 0.006) and lower eGFR (72.83 vs. 80.99 mL/min/1.73 m^2^, *p* = 0.036). Patients with moderate to severe F-MR significantly differed in LA diameter (42.9 vs. 40.00 mm, *p* = 0.016), and tricuspid regurgitation frequency from the group without F-MR or with trace or mild F-MR was observed both in TTE (12.5% vs. 2.68%, *p* = 0.033) and TEE (29.17% vs. 9.01% *p* = 0.007). Moderate to severe F-MR did not affect three-month PVI efficacy, but it seemed to modify the long-term efficacy of the procedure (*p* = 0.041) (Table 2).

### 3.4. Three-Month PVI Efficacy

PVI efficacy assessed three months after the procedure reached 75.7%.

Patients with successful PVI evaluated three months after the procedure did not differ from the ineffective group in the majority of the clinical characteristics. Only highly symptomatic AF (EHRA 4, *p* = 0.003) and hypothyreosis (*p* = 0.019) predisposed patients to the unsuccessful effect of the procedure (Table 3).

No significant difference in TTE parameters was observed depending on the PVI success. In TEE, only severe F-MR had a significant effect on three-month PVI efficacy (*p* = 0.012).

In stepwise logistic regression, three-month PVI efficacy was associated with the presence of HFpEF (odds ratio (OR): 0.12, 95% confidence interval (CI): 0.02–0.65; *p* = 0.014), EHRA 4 (OR: 0.03, 95% CI: 0.003–0.4; *p* = 0.008), hypothyroidism (OR: 0.2, 95% CI: 0.07–0.73; *p* = 0.01), and RWT (OR: 101 × 10^3^, 95% CI: 14.24–712 × 10^6^; *p*= 0.01).

### 3.5. Long-Term PVI Efficacy

PVI efficacy assessed three months after the procedure reached 75.7%, but 36% of patients experienced arrhythmia recurrence (mean time to recurrence: 9.4 ± 5.9 months).

Individuals with sinus rhythm maintenance did not significantly differ in anthropometric parameters from patients without long-term PVI success. Although obesity was diagnosed in 42.5% of patients with long-term unsuccessful PVI and only in 26.5% of individuals without AF, the difference did not reach statistical significance (*p* = 0.06) (Table 4).

Patients with long-term unsuccessful PVI did not differ in terms of the majority of clinical characteristics from the control group, but the most symptomatic patients (EHRA 4) experienced AF recurrence more frequently (*p* = 0.037).

Significant differences in terms of echocardiographic parameters measured in TTE were not observed. In TEE, moderate to severe AF-MR was more frequently observed in patients with unsuccessful PVI assessed in long-term follow-up (*p* = 0.041).

The receiver operating characteristic (ROC) analysis showed no association of CHA2DS2-VASc score and echocardiographic parameters (LA, LAA, TTE F-MR grade, and TEE F-MR grade) with the risk of AF recurrence in long-term follow-up.

In the Kaplan–Meier analysis, only moderate to severe F-MR diagnosed in TEE had an impact on the long-term procedure outcome (*p* = 0.048) (Figure 1).

## 4. Discussion

In our research, F-MR was observed in most of the patients qualified for PVI, independently from the used echocardiographic technique. While trace and mild F-MR were the most commonly observed, only significant F-MR predisposed patients to an unsuccessful PVI outcome.

In the study by Yasuda et al. [6] that was designed to identify the predictors of successful catheter ablation for AF, PVI efficacy after two sessions of ablation reached 75%. The authors reported a higher incidence of paroxysmal AF in individuals with the successful procedure (paroxysmal: 96% vs. persistent: 46%, *p* < 0.001). Although the PVI effectiveness rate complies with our study results, in our research, no significant difference in terms of ablation efficacy between patients with a paroxysmal and persistent AF was observed. In the cited study, successful ablation was related to the lower incidence of F-MR in TTE (10% vs. 46%, *p* < 0.01). That observation does not correspond with our research results, which could be due to the influence of differences in the study group characteristics. In our study, three-month PVI efficacy was not associated with the presence of F-MR of any grade in TTE and TEE. Only severe F-MR observed in TEE had an influence on the PVI outcome.

Patients with F-MR are at risk of ablation inefficacy, and are probably predisposed to experiencing recurrent arrhythmia after sinus rhythm restoration. On the other hand, a successful AF ablation procedure could improve F-MR severity through LA reverse remodeling [7].

According to Zhao et al. [8], F-MR, but also AF duration and LA diameter, were independent predictors of the arrhythmia recurrence in long-term observation after catheter ablation. Furthermore, the grade of F-MR was associated with the AF recurrence rate. In our research, moderate to severe F-MR measured in TEE, but not in TTE, was more frequently observed in patients with AF recurrence.

Similarly, in the study by Gertz et al. [9], F-MR was associated with the AF recurrence rate in a long-term follow-up. Individuals with F-MR were characterized by a greater LA diameter (45 vs. 41 mm, *p* < 0.0001) and higher incidence of persistent AF (71 vs. 28%, *p* < 0.0001). F-MR patients compared to the group without significant F-MR suffered from AF recurrence more often (61 vs. 46%, *p* = 0.04). The risk of AF recurrence was not only associated with LA size, but also with the degree of F-MR expressed as the F-MR/LA ratio (0.25 vs. 0.2, *p* = 0.03).

In the study by Qiao et al. [10], AF-MR assessed in TTE was strongly associated with atrial substrate remodeling, as well as with the AF recurrence risk in long-term observation after catheter ablation. Low voltage zones observed in the mapping of LA were present in 64.9% patients with F-MR and only in 22.1% of individuals without mitral valve disease (*p* < 0.001). The study revealed that F-MR constituted an independent predictor of low voltage zone presence (OR = 7.286; 95% CI = 3.023–17.562; *p* < 0.001). In the follow-up, 60% of patients with F-MR experienced AF recurrence, compared to 19.5% in the group without F-MR. In a multivariate analysis, F-MR was an independent predictor of AF recurrence (HR 2.291; 95% CI = 1.062–4.942; *p* = 0.03).

F-MR in patients with AF is considered to be the trigger of symptoms related to heart failure. Therefore, it represents an incentive therapeutic target in AF patients and HFpEF. It should be noted that successful PVI may lead to reverse atrial remodeling and presumably could reduce the severity of F-MR.

In the study by Gertz et al. [11], patients with at least moderate F-MR were compared to the control group. The F-MR group was characterized by older age and higher frequency of persistent FA (62 vs. 23%, *p* < 0.0001). In TTE, AF-MR was related to the LA volume index (32 vs. 26 cm^3^/m^2^, *p* = 0.008) and mitral valve annular size (34.9 mm vs. 32.3 mm, *p* = 0.001).

In TTE assessed during follow-up, patients maintaining sinus rhythm had greater reductions in LA volume index and mitral valve annular size. Significant F-MR occurred less often in individuals without AF recurrence (24% vs. 82%, *p* = 0.005).

In the study by Nishino et al. [12] comprising 43 patients who underwent successful ablation for persistent AF, significant reverse remodeling of MV apparatus and LA was observed. After maintaining sinus rhythm for six months, not only did the annular area decrease (5.32 ± 0.9 vs. 4.73 ± 0.8 cm^2^/m^2^; *p* < 0.001), but MA contraction also recovered (7.51% vs. 9.71%; *p* = 0.008). The leaflet surface area significantly decreased (5.74 vs. 5.19 cm^2^/m^2^; *p* < 0.001), without changes in tenting volume and height. The improvement in AF MR grade expressed as a F-MR jet area was noticed in the predominance of patients (1.83 vs. 0.77 cm^2^; *p* < 0.001).

The limited number of enrolled patients combined with the possible measurement error in echocardiography and consideration of both paroxysmal and persistent AF were the main limitations of the study. F-MR grade assessment based on EACVI guidelines included only qualitative and semi-quantitative parameters, quantitative values were not obtained. The MV E/A ratio was the only analyzed diastolic parameter. While persistent AF was not the exclusion criterion in the study, the possible impact of the presence of AF on F-MR grade was not analyzed.

Furthermore, echocardiographic evaluation at the follow-up was not performed, which resulted in an impossibility of MV remodeling assessment.

Although the evaluation of low voltage areas in patients with a significant MR could be valuable, data from 3D-electroanatomical mapping were not analyzed. The incidence of non-pulmonary vein trigger AF, which could be a significant problem in patients with advanced MR, was not assessed.

Although PVI efficacy assessment without performing seven-day Holter monitoring is not flawless, patients enrolled to the study were highly symptomatic before the procedure. PVI efficacy based on 12-lead ECG and self-assessment in this group of patients seems to be reliable.

## 5. Conclusions

F-MR is observed in most of the patients with preserved EF qualified to PVI. Trace and mild F-MR are most often diagnosed, but only significant F-MR in TEE has an impact on three-month and long-term PVI efficacy. Because TEE is performed in every patient qualified for PVI to exclude the presence of LA appendage trombi, the F-MR assessment in TEE could be used to predict the procedure efficacy. Our study results establish the pivotal role of this imaging technique in the process of the qualification of AF ablation.

Patients with significant F-MR in TEE are burdened with a risk of unsuccessful PVI. This group needs regular follow-ups. In some patients, an invasive valvular treatment, including cardio surgery procedures (mitral valve plasty, surgical ablation) should be taken under consideration to improve MV function and reduce AF recurrence risk.

## Figures and Tables

**Figure 1 medicina-57-00798-f001:**
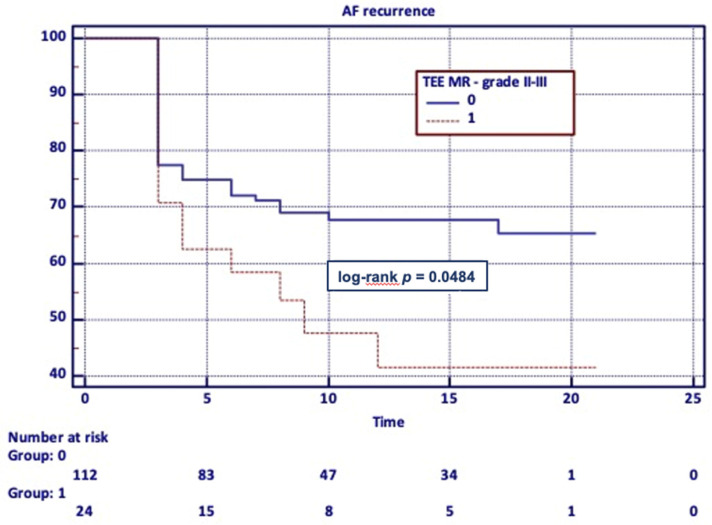
Long-term PVI efficacy—Kaplan–Meier analysis of F-MR severity.

**Table 1 medicina-57-00798-t001:** Functional mitral regurgitation (F-MR) in patients who qualified for PVI, assessed in transthoracic (TTE) and transesophageal echocardiography (TEE).

Variable	Absolute Count and Percentage	
	TTE	TEE
F-MR any grade	101 (74.3%)	129 (94.9%)
F-MR trace	36 (26.5%)	24 (17.6%)
F-MR mild	59 (43.4%)	81 (59.6%)
F-MR moderate	5 (3.7%)	22 (16.2%)
F-MR severe	1 (0.7%)	2 (1.5%)

Abbreviations: F-MR, functional mitral regurgitation; TTE, transthoracic echocardiography; TEE, transesophageal echocardiography.

**Table 2 medicina-57-00798-t002:** Clinical characteristics and echocardiographic parameters depending on presence of moderate to severe F-MR in TEE.

Variable	Absolute Count and Percentage or Median and 25–75 Percentile or Mean ± Standard Deviation		*p*
	Moderate to Severe F-MR		
	No *n* = 112	Yes *n* = 24	
Age (years)	58.6 (11.0)	62 (9.1)	0.102
Sex (F/M)	41/71	6/18	0.278
Weight (kg)	85.29 (14.6)	86.69 (18.6)	0.857
Height (cm)	172.56 (9.7)	171.17 (8.9)	0.587
BMI (kg/cm^2^)	28.57 (3.9)	29.37 (4.7)	0.423
Paroxysmal AF	92 (82.1%)	18 (75.0%)	0.728
Persistent AF	17 (15.8%)	5 (20.8%)	0.495
Long-standing persistent AF	3 (2.7%)	1 (4.2%)	0.695
CHA2DS2-VASc	2 (1–3)	2 (1–3)	0.822
EHRA			
1	2 (1.8%)	0 (0%)	0.510
2	54 (48.2%)	13 (54.2%)	0.597
2a	5 (4.5%)	2 (8.3%)	0.436
2b	49 (43.8%)	11 (45.8%)	0.852
3	52 (46.4%)	10 (41.7%)	0.671
4	4 (3.6%)	1 (4.2%)	0.088
HFpEF	4 (4.6%)	4 (8.2%)	0.396
Echocardiographic parameters			
LV EF (%)	56.13 (3.9)	55.08 (3.5)	0.181
LA (mm)	40.00 (4.5)	42.96 (5.5)	**0.016**
LAA (cm^2^)	21.59 (4.3)	23.27 (5.0)	0.136
MV E/A	1.14 (0.5)	1.16 (0.6)	0.976
LV EDD (mm)	50.85 (5.3)	52.71 (5.1)	0.092
LV ESD (mm)	30.40 (5.3)	31.25 (4.5)	0.350
RWT	0.38 (0.08)	0.37 (0.1)	0.675
LV mass index (g/m^2^)	103.91 (30.1)	111.23 (24.2)	0.097
Creatinine (mg/dL)	0.91 (0.21)	1.06 (0.25)	**0.006**
eGFR (mL/min/1.73 m^2^)	80.99 (12.58)	72.83 (17.25)	0.036
TR mild in TTE	3 (2.68%)	3 (12.5%)	**0.033**
TR mild in TEE	10 (9.01)	7 (29.17)	**0.007**
3-month PVI efficacy	86 (76.8%)	17 (70.8%)	0.537
Long-term PVI efficacy	36 (32.1%)	13 (54.2%)	**0.041**

Abbreviations: F-MR, functional mitral regurgitation; BMI, body mass index; AF, atrial fibrillation; HFpEF, heart failure preserved ejection fraction; LV EF, left ventricle ejection fraction; LA, left atrium; LAA, left atrium area; MV E/A, mitral valve E wave to A wave ratio; LV EDD, left ventricle end-diastolic diameter; LV ESD, left ventricle end-systolic diameter; RWT, relative wall thickness; LV, left ventricle; eGFR, estimated glomerular filtration rate; TTE, transthoracic echocardiography; TEE, transesophageal echocardiography; TR tricuspid regurgitation; PVI, pulmonary vein isolation. bold indicates significant value.

**Table 3 medicina-57-00798-t003:** Clinical characteristics and echocardiographic parameters depending on PVI efficacy assessed three months after the procedure.

Variable	Absolute Count and Percentage or Median and 25–75 Percentile or Mean ± Standard Deviation		*p*
	Three-Month PVI Efficiency		
	No *n* = 33	Yes *n* = 103	
Age (years)	60.15 (8.7)	59.02 (11.4)	0.945
Sex (F/M)	11/22	36/67	0.865
Weight (kg)	85.85 (15.2)	85.43 (15.4)	0.887
Height (cm)	173.24 (8.9)	172.02 (9.8)	0.388
BMI (kg/cm^2^)	28.48 (3.8)	28.79 (4.2)	0.726
Paroxysmal AF	27 (81.8%)	80 (80.6%)	0.728
Persistent AF	6 (18.2%)	16 (15.6%)	0.719
Long-standing persistent AF	0 (0.0%)	4 (3.9%)	0.251
CHA2DS2-VASc	2 (1–3)	2 (1–3)	0.619
EHRA			
1	0 (0.0%)	2 (1.9%)	0.420
2	16 (48.9%)	51 (49.5%)	0.918
2a	2 (6.1%)	5 (4.9%)	0.785
2b	14 (42.4%)	46 (44.7%)	0.822
3	13 (39.4%)	49 (47.6%)	0.412
4	4 (12.2%)	1 (0.9%)	**0.003**
HFpEF	4 (12.2%)	4 (3.8%)	0.08
Echocardiographic parameters			
LV EF (%)	56.39 (5.3)	55.81 (3.6)	0.374
LA (mm)	40.45 (5.4)	40.54 (4.7)	0.807
LAA (cm^2^)	22.07 (4.3)	21.84 (4.5)	0.949
MV E/A	1.09 (0.4)	1.16 (0.5)	0.738
LV EDD (mm)	51.67 (5.43)	51.02 (5.3)	0.470
LV ESD (mm)	31.36 (5.2)	31.3 (5.1)	0.246
RWT	0.36 (0.06)	0.39 (0.1)	0.086
LV mass index (g/m^2^)	101.21 (24.1)	106.48 (30.7)	0.352
TTE F-MR any grade	24 (72.7%)	77(74.8%)	0.816
TTE F-MR trace	9 (27.3%)	27 (26.2%)	0.904
TTE F-MR mild	13 (39.4%)	46 (44.7%)	0.595
TTE F-MR moderate	1 (3.0%)	4 (3.9%)	0.821
TTE F-MR severe	1 (3.0%)	0 (0.0%)	0.076
TTE F-MR moderate-severe	2 (6.1%)	4 (3.9%)	0.596
TEE F-MR any grade	31 (93.9%)	98 (95.6%)	0.785
TEE F-MR trace	8 (24.2%)	16 (15.5%)	0.253
TEE F-MR mild	16 (48.5%)	65 (63.1%)	0.253
TEE F-MR moderate	5 (15.2%)	17 (16.5%)	0.854
TEE F-MR severe	2 (6.0%)	0 (0.0%)	**0.012**
TEE F-MR moderate-severe	7 (21.2%)	17 (16.5%)	0.537

Abbreviations: F-MR, functional mitral regurgitation; BMI, body mass index; AF, atrial fibrillation; HFpEF, heart failure preserved ejection fraction; LV EF, left ventricle ejection fraction; LA, left atrium; LAA, left atrium area; MV E/A, mitral valve E wave to A wave ratio; LV EDD, left ventricle end-diastolic diameter; LV ESD, left ventricle end-systolic diameter; RWT, relative wall thickness; LV, left ventricle; PVI, pulmonary vein isolation; TTE, transthoracic echocardiography; TEE, transesophageal echocardiography. bold indicates significant value.

**Table 4 medicina-57-00798-t004:** Clinical characteristics and echocardiographic parameters depending on long-term PVI efficacy.

Variable	Absolute Count and Percentage or Median and 25–75 Percentile or Mean ± Standard Deviation		*p*
	Long-Term PVI Efficacy		
	Yes *n* = 87	No *n* = 49	
Age (years)	58.0 (11.8)	61 (8.5)	0.399
Sex (F/M)	29/58	18/31	0.689
Weight (kg)	86.27 (14.9)	84.22 (16)	0.457
Height (cm)	172.14 (9.4)	172.63 (9.9)	0.773
BMI (kg/cm^2^)	29.07 (4.3)	28.08 (3.6)	0.176
Paroxysmal AF	70 (80.4%)	40 (81.6%)	0.728
Persistent AF	14 (16%)	8 (16.3%)	0.972
Long-standing persistent AF	3 (3,5%)	1 (2%)	0.641
CHA2DS2-VASc	2 (1–3)	2 (1–3)	0.926
EHRA			
1	2 (2.3%)	0 (0%)	0.285
2	44 (50.6%)	23 (46.9%)	0.684
2a	5 (5.7%)	2 (4%)	0.673
2b	39 (44.8%)	21 (42.9%)	0.824
3	40 (45.9%)	22 (44.9%)	0.903
4	1 (1.2%)	4 (8.2%)	**0.037**
HFpEF	4 (4.6%)	4 (8.2%)	0.396
Echocardiographic parameters			
LV EF (%)	55.80 (3.8)	56.20 (3.9)	0.427
LA (mm)	40.66 (4.9)	40.29 (4.9)	0.797
LAA (cm^2^)	22.19 (4.5)	21.38 (4.3)	0.307
MV E/A	1.2 (0.5)	1.05 (0.5)	0.123
LV EDD (mm)	51.08 (5.4)	51.35 (5.1)	0.691
LV ESD (mm)	30.30 (5.2)	31 (5.0)	0.344
RWT	0.39 (0.09)	0.36 (0.1)	0.074
LV mass index (g/m^2^)	106.82 (32.1)	102.33 (23.4)	0.515
TTE F-MR any grade	61 (70.1%)	40 (81.6%)	0.140
TTE F-MR trace	23 (26.4%)	13 (26.5%)	0.990
TTE F-MR mild	35 (40.2%)	24 (48.9%)	0.323
TTE F-MR moderate	3 (3.5%)	2 (4.1%)	0.851
TTE F-MR severe	0 (0.0%)	1 (2.04%)	0.181
TTE F-MR moderate-severe	3 (3.5%)	3 (6.1%)	0.466
TEE F-MR any grade	82 (94.3%)	47 (95.9%)	0.673
TEE F-MR trace	16 (18.4%)	8 (16.3%)	0.762
TTE F-MR mild	55 (63.2%)	26 (53.0%)	0.242
TTE F-MR moderate	11 (12.6%)	11 (22.5%)	0.136
TTE F-MR severe	0 (0%)	2 (4.1%)	0.058
TTE F-MR moderate-severe	11 (12.6%)	13 (26.5%)	**0.041**

Abbreviations: F-MR, functional mitral regurgitation; BMI, body mass index; AF, atrial fibrillation; HFpEF, heart failure preserved ejection fraction; LV EF, left ventricle ejection fraction; LA, left atrium; LAA, left atrium area; MV E/A, mitral valve E wave to A wave ratio; LV EDD, left ventricle end-diastolic diameter; LV ESD, left ventricle end-systolic diameter; RWT, relative wall thickness; LV, left ventricle; PVI, pulmonary vein isolation; TTE, transthoracic echocardiography; TEE, transesophageal echocardiography. bold indicates significant value.

## Data Availability

The data presented in this study are available on request from the corresponding author.

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
