# Peer review of "Functional Mitral Regurgitation in Patients with Preserved Ejection Fraction Qualified for Pulmonary Vein Isolation: A Negative Prognostic Factor for Catheter Ablation Efficacy"

_medicina, 2021, doi:10.3390/medicina57080798_

Round 1

Reviewer 1 Report

I revised Cichon's manuscript, which analyzes the association between atrial fibrillation ablation efficacy and AF-MR. While reading the text, I noticed the creating conflicts:

Patients with atrial functional mitral regurgitation have a normal mitral valve, a dilated mitral annulus, and no left ventricular remodeling. However, in the study of Cichon et al. 50% of patients had LV remodeling or LV hypertrophy. As a result, I feel the manuscript's title and terms (AF-MR) should be modified.

According to echocardiographic investigations (Zhou et al. Impact of atrial fibrillation on tricuspid and mitral annular dilatation and valvular regurgitation. Otsujo et al. Isolated annular dilation does not usually cause important functional mitral regurgitation: comparison between patients with lone atrial fibrillation and those with idiopathic or ischemic cardiomyopathy) in patients with lone atrial fibrillation and normal LV size and function, isolated mitral annular dilatation does not result in substantial mitral regurgitation. It's difficult to imagine that the only cause of severe mitral regurgitation in the two patients with severe AF-MR in the Cichon et al. group was mitral annulus dilatation. The authors implicitly indicate that atrial functional regurgitation causes substantial mitral regurgitation.

The following studies provide compelling evidence for an indirect association between valve dilation and MR rather than a causal link: Glower et al. Pure annular dilation as a cause of mitral regurgitation: a clinically distinct entity of female heart disease. Tanimoto et al. Effect of isolated left atrial enlargement on mitral annular size and valve competence. Kihara et al. Mitral regurgitation associated with mitral annular dilation in patients with lone atrial fibrillation: an echocardiographic study

Author Response

Thank You for Your insight. AF MR is a relatively new issue, still constituting a subject to research. However, new Carpentier Classification presented in 2020 in JACC takes AF MR into account (Carpentier I type with normal leaflet morphology).

As You have mentioned, according to the definitions AF MR is a functional MR observed in patients with normal systolic LV function. However, in AF MR remodeling processes concern not only LA, but also LV. Although EF reaches normal values, global longitudinal strain may be impaired. 

AF MR typically occurs in the context of AF and HFpEF with severe LA dilation. Long lasting AF results in LV diastolic dysfunction and HFpEF development, as well as in LA dilation leading to AF MR. On the other hand, HFpEF could be a trigger for AF, but also could increase LA pressure, which results in LA dilation an AF MR.

All presented pathophysiological dependance lead to the fact that typically patients with AF MR are characterized with a borderline EF (50%) and diastolic dysfuntion but also with some degree of structural LV remodeling. Results of our study are compatible with these findings.

What’s more according to the results presented this year in General Thoracic and Cardiovascular Surgery [(2021) 69:1041–1049] chronic AF at early stage leads to mild- moderate mitral, as well as tricuspid regurgitation. In case of long lasting arrythmia extended annular dilation could result in severe mitral and tricuspid regurgitation, but also in LV and RV dilation. These results could explain presence of severe MR in a small group of patients in our study, probably due to long lasting AF.

Reviewer 2 Report

Reviewer comment:

In this retrospective study, Cichoń M et al. sought to determine the impact of AF-MR on pulmonary vein isolation (PVI) efficacy. The authors found that severe AF-MR in TEE at baseline was associated with lower 3-month PVI efficacy and moderate to severe AF-MR in TEE was related to inefficient PVI assesed in long-term follow-up. Although this study may contain important clinical implications, there exist several critical limitations.

Major concerns:

#1.         It was not surprising that patients with AF-MR were at risk of recurrence after catheter ablation, which was in line with previous literature. As the authors stated in the discussion, maintenance of sinus rhythm might have been associated with the improvement in AF-MR severity because of gradual reverse remodeling of the left atrium after catheter ablation during sinus rhythm. Patients with improved AF-MR may have better outcome than those without AF-MR, while those with persistent moderate to severe AF-MR may have had higher recurrence rate. Therefore, it is fair to compare 12-month outcomes between patients with mild AF-MR, those with moderate to severe AF-MR with improvement at 3 months, and those with moderate to severe AF-MR without improvement at 3 months to assess the impact of the improvement of AF-MR during an early phase following ablation procedures.

#2.         Assessment of MR severity greatly depends on the rhythm status during TTE or TEE. Due to the lack of atrio-ventricular synchronized organized conduction and irregular rhythm, severity of MR during AF may be worse than that during sinus rhythm. The authors should clarify the rhythm status during echocardiography and include only the patients undergoing echocardiography during sinus rhythm.

#3.         Atrial fibrosis is a known risk factor for AF recurrence after catheter ablation. Considering that patients with moderate or severe MR had larger LA than that those without, high degree of atrial fibrosis represented by the extent of low voltage areas using 3-D electroanatomical mapping would be found in those with advanced AF-MR. It is worth investigating the association between the severity of AF-MR degree and the extent of low voltage areas on 3D electroanatomical mapping.

#4.         Patients with advanced AF-MR may have higher incidence of non-PV trigger AF than those without. How was the incidence of non-PV trigger AF during the ablation procedures in patients with advanced AF-MR in comparison with those without?

#5.         In Figure 1, as the Kaplan-Meier analysis only moderate to severe AF-MR diagnosed in TEE had impact on long-term outcome, other figures are unnecessary.

Minor concerns:

#1.         In the result section, the authors described the difference in serum creatinine concentration, eGFR, and tricuspid regurgitation frequency among the patients with different AF-MR grades, which should be summarized in Table 2.

#2.         In ‘3-month PV efficacy’ of the results, the authors described as follows ‘Patients with successful PVI evaluated after 3 months after the procedure’, which should be ‘evaluated 3 months after the procedure’ or ‘evaluated after 3 months following the procedure’.

Author Response

Major concerns:

#1.         It was not surprising that patients with AF-MR were at risk of recurrence after catheter ablation, which was in line with previous literature. As the authors stated in the discussion, maintenance of sinus rhythm might have been associated with the improvement in AF-MR severity because of gradual reverse remodeling of the left atrium after catheter ablation during sinus rhythm. Patients with improved AF-MR may have better outcome than those without AF-MR, while those with persistent moderate to severe AF-MR may have had higher recurrence rate. Therefore, it is fair to compare 12-month outcomes between patients with mild AF-MR, those with moderate to severe AF-MR with improvement at 3 months, and those with moderate to severe AF-MR without improvement at 3 months to assess the impact of the improvement of AF-MR during an early phase following ablation procedures.

 Thank You for rising that problem. We agree that lack of echocardiographic evaluation at the follow- up constitute a weak point of the study. As it resulted in impossibility of MV remodeling assessment, we decided to take that problem into account in Limitations.

#2.         Assessment of MR severity greatly depends on the rhythm status during TTE or TEE. Due to the lack of atrio-ventricular synchronized organized conduction and irregular rhythm, severity of MR during AF may be worse than that during sinus rhythm. The authors should clarify the rhythm status during echocardiography and include only the patients undergoing echocardiography during sinus rhythm.

Thank You for that comment. We had clarified the rhythm during TEE before PVI. Most of the patients qualified to the study had sinus rhythm during PVI- in group without MR or with MR trace-mild: 86 (76%) and in MR moderate-severe 18 (75%). AF was observed in 26 patients (23%) without MR or with MR trace-mild and in 6 patients (25%) in MR moderate-severe. The groups did not differ in AF frequency (P=0.48). It does not change the fact, that excluding patients with AF would improve the qualities of the study. We will take it into account in our next projects.

#3.         Atrial fibrosis is a known risk factor for AF recurrence after catheter ablation. Considering that patients with moderate or severe MR had larger LA than that those without, high degree of atrial fibrosis represented by the extent of low voltage areas using 3-D electroanatomical mapping would be found in those with advanced AF-MR. It is worth investigating the association between the severity of AF-MR degree and the extent of low voltage areas on 3D electroanatomical mapping.

 Thank You for Your insight. Exploration of low voltage areas in patients with MR using 3D electroanatomical mapping is an interesting concept for further investigation.  The purpose of the study was to assess usefulness of commonly used techniques (TTE, TEE) in PVI efficacy prediction so, 3D mapping results were not included.

#4.         Patients with advanced AF-MR may have higher incidence of non-PV trigger AF than those without. How was the incidence of non-PV trigger AF during the ablation procedures in patients with advanced AF-MR in comparison with those without?

Thank You for rising that problem. As we said the focal point of the study was echocardiography and information from 3D electroanatomical mapping were not collected. The problem of non-PV trigger AF could be interesting subject of next studies, as our department performs a large number of PVI and we would be able to collect material according to Yours suggestions.  

#5.         In Figure 1, as the Kaplan-Meier analysis only moderate to severe AF-MR diagnosed in TEE had impact on long-term outcome, other figures are unnecessary.

According to Yours suggestion we modified Figure 1.

Minor concerns: 

#1.         In the result section, the authors described the difference in serum creatinine concentration, eGFR, and tricuspid regurgitation frequency among the patients with different AF-MR grades, which should be summarized in Table 2.

Thank You for Your suggestions. We had added lines in Table 2 to summarize that information. 

#2.         In ‘3-month PV efficacy’ of the results, the authors described as follows ‘Patients with successful PVI evaluated after 3 months after the procedure’, which should be ‘evaluated 3 months after the procedure’ or ‘evaluated after 3 months following the procedure’.

Thank You for Your insight. We changed the sentence into “evaluated 3 months after the procedure”.

Round 2

Reviewer 1 Report

The authors did not respond to my inquiries on  the 50%  remodeling and hypertrophy of the left ventricle, which defies the main point of the study.

In these circumstances, the article has a significant flaw. And if it isn't fixed, the manuscript can't be released in its current state.

Author Response

According to Yours suggestions we decided to change “Atrial functional mitral regurgitation” into “Functional mitral regurgitation” highlighting at the same time, that the study refers to patients with preserved EF. We also had included information, that MR in the study group could be the result of LA, but also LV remodeling. All corrections in text are marked in orange. Thank You for Your insight.

Reviewer 2 Report

Thank you for the opportunity to review the revised manuscript. Although the authors responded the comments, the manuscript has not been sufficiently revised according to the comments, which requires further revision to give a sufficiently high priority rating to warrant acceptance for publication in the journal.

Author Response

According to Yours suggestions we decided to add some comments in our manuscript. All the changes in text are marked in orange.

#1 We decided to take that problem into account in Limitations (Line 323).

#2 Thank You once again for that suggestion. We agree, that excluding patients with AF during TEE would improve the qualities of the study. We added information in Limitations (Line 324-325).

#3 and  #4 We must admit, that 3D mapping results were not included, as the purpose of the study was to assess usefulness of commonly used techniques (TTE, TEE) in PVI efficacy prediction (Lines 329-330 and 330-332).